# Emergency Department Visits among Cancer Patients during SARS-CoV-2 Pandemic

**DOI:** 10.3390/cancers15041240

**Published:** 2023-02-15

**Authors:** Davide Valsecchi, Luca Porcu, Abdelrahman Khater, Rosa Alessia Battista, Leone Giordano, Stefano Cascinu, Andrea Assanelli, Chiara Lazzari, Vanesa Gregorc, Aurora Mirabile

**Affiliations:** 1Emergency Department, IRCCS San Raffaele Scientific Institute, 20132 Milan, Italy; 2Methodology for Clinical Research Laboratory, Oncology Department, Istituto di Ricerche Farmacologiche Mario Negri IRCCS, 20156 Milan, Italy; 3Joslin Diabetes Center, Harvard Medical School, Boston, MA 02215, USA; 4Department of Otorhinolaryngology, IRCCS San Raffaele Scientific Institute, 20132 Milan, Italy; 5Department of Medical Oncology, IRCCS San Raffaele Scientific Institute, 20132 Milan, Italy; 6Medical Oncology Department, Candiolo Cancer Institute, FPO-IRCCS, 10060 Turin, Italy

**Keywords:** SARS-CoV-2 or COVID-19, cancer patients, emergency department

## Abstract

**Simple Summary:**

During COVID 19 pandemic, cancer patients, their caregivers and physicians needed to balance the challenges associated with pandemic ensuring cancer care. In this paper, we analysed ED visits during the 72 days of the pandemic in 2020 (Italian lockdown period) and compared them to the ED visits in the same calendar days in 2019 and 2021. We compared their severity, outcome (admission vs. discharge vs. death vs. hospice/palliative care), method of arrival to the ED and type of tumours affecting patients, suggesting that pandemic related emotional distress and hospital departmental reorganization could have nega-tively influenced ED admissions. Our aim was to highlight how much the government restrictive measures could have had an impact on emergency care for fragile patients such as cancer patients and help to understand how to reconcile the health needs of a specific class of patients with the need to protect public health.

**Abstract:**

The coronavirus disease 2019 (COVID-19) pandemic has had a global impact. Patients with cancer, their caregivers, and physicians need to balance the challenges associated with COVID-19 while ensuring cancer care. Nevertheless, emotional distress and hospital departmental reorganization could have led to a decrease in ED admissions even among oncological patients. Methods: We compared the 72 days of the pandemic in 2020 with the same calendar days in 2019 and 2021, defining a 20% decrease in ED visits as clinically significant. We studied the cause for visit, its severity, outcome (admission vs. discharge vs. death vs. hospice/palliative care), the tumor site, and method of arrival to the ED for the 3 time periods. Results: A significant decrease in ED oncological visits was found in 2020 compared to 2019, before returning to similar numbers in 2021. Fear, anxiety, and worry, in addition to hospital departmental reorganization, surely had an important role in the delay of ED visits, which resulted in irreparable consequences.

## 1. Introduction

The coronavirus disease 2019 (COVID-19) pandemic has posed a serious public health concern since its first reported outbreak in China late in 2019, before spreading worldwide in the early months of 2020. The SARS-CoV-2 virus is known to spread mostly via respiratory droplets; however, transmission through fomites from infected patients or the environment is also possible [1,2].

Compared to the general population, oncologic patients are at a higher risk of worse outcomes due to infectious diseases [3,4]. This remained true during the COVID-19 pandemic, as they were reported to be at increased risk of infection with the SARS-CoV-2 virus and of developing a more severe disease course. In particular, a large proportion of this sub-population necessitated high levels of intensive care, experienced a more rapidly evolving disease, and carried a higher mortality risk [5].

Moreover, cancer patients require an uninterrupted care pathway, as most of their diagnostic and therapeutic interventions are essential and time sensitive. At the same time, COVID-19 exposure could become risky or even fatal for these patients [6].

In February 2020, the first positive cases in Italy arising from locally acquired infections were reported [7]. In the last week of February 2020 in Lombardy, 531 patients were reported positive, which rose to 2612 by 7 March 2020. Consequently, decrees issued by the Prime Minister of Italy put the country in an eventual lockdown, aiming to slow or potentially halt the spread of SARS-CoV-2 [8,9,10]. This, in turn, created difficulties for patients trying to reach our hospital, as public transportation access and schedule frequencies were reduced, and many people had to be placed under a 2-week self-quarantine based on exposure risk [11].

Cancer patients need regular access to healthcare for life-sustaining treatments, where delays can be detrimental [12]. Meanwhile, many cancer patients are also immunocompromised and may have worse outcomes due to COVID-19, should they get infected while seeking treatment [13]. In view of these competing concerns, patients with cancer are forced to choose between seeking oncologic treatment and increasing the risk of contracting COVID-19 or postponing therapy and minimizing the risk of contracting COVID-19 [14].

Another unforeseen challenge caused by the pandemic is the dramatic reduction in social contact through isolation and distancing measures. This can be damaging to oncologic patients’ well-being as they rely on social support to get through an already difficult time [15]. The presence of a support network helps reduce psychological symptoms in these patients [16] and reduces the risk of morbidity and mortality [17]. 

Many patients fear dying alone; thus, having family members and friends physically present becomes a crucial aspect of providing patient-centered care [18]. These needs were difficult to meet, however, given that social distancing was a key policy in managing this COVID-19 pandemic [19]. At the same time, many patients showed an understanding of their increased risk of severe complications in the case of infection [20]. One potential consequence of this emotional distress is a decrease in Emergency Department (ED) visits among oncologic patients, which, to our knowledge, has not been reported in the literature except for an abstract about Philadelphia hospitals, published by Zachary et al. in JCO 2020 [21].

This study aimed to understand and document how the pandemic affected ED admissions in oncological patients.

## 2. Materials and Methods

This was a retrospective cross-sectional study in a tertiary center of national relevance with the primary aim to statistically detect and estimate the change in oncologic patients’ ED visits during the lockdown period in 2020 and again during the same period of the following year. The primary endpoint was the number of oncological patients’ ED visits during the 2020 lockdown period in Lombardy, compared to the same days in 2019 (pre-pandemic) and in 2021 (no lockdown and established measures in place such as vaccinations). The observation period was 72 days for each of the three years. The observation period was determined before data collection took place.

The secondary aim was to compare the different time periods with regards to the reasons for visit, severity of the visit, outcome (admission vs. discharge vs. death vs. hospice/palliative care) of the visit, site of tumor, and method of access to the visit.

All oncologic patients that accessed the ED in the defined time periods were included, except for adults < 18 years of age, patients incapable of giving informed consent, and patients who completed their cancer treatments more than 5 years before the date of the study.

The present study was performed in accordance with the Declaration of Helsinki (6th revision, 2008), and the study protocol was reviewed and approved by the local ethics committee.

### Statistical Analysis

For each observation period, the number of oncological patients’ ED visits was calculated, overall and by week. Odds were used to estimate the chance of oncological patients’ ED visits in one year (i.e., 2020 or 2021) compared to the chance of oncological patients’ ED visits in the reference year (i.e., 2019). Odds were also used to estimate the change in oncological patients’ visits to the ED in one year (i.e., 2020 or 2021) compared to the reference year (i.e., 2019). The exact binomial test was used to formally compare previous probabilities. Univariable logistic regression models were used to detect and estimate the statistical association between the type of ED visits/patients’ characteristics and the year of oncological patients’ ED visits (i.e., 2020 vs. 2019 and 2021 vs. 2019). 

Multivariable logistic regression models were used to identify statistically independent factors. Statistical analysis was generated using SAS software for Windows, version 9.4 (SAS Institute Inc., Cary, NC, USA; 2016).

## 3. Results

Between 7 March 2019 and 17 May 2019, a total of 15,864 ED visits were recorded, of which 1061 (6.7%) were oncologic patients. In the same period in 2020, a total of 6233 ED visits were recorded, of which 459 (7.4%) were oncologic patients. Meanwhile, in 2021, a total of 10,352 ED visits were observed, of which 996 (9.6%) were oncologic patients.

A statistically significant decrease was found for total ED visits (odds = 0.39, *p* < 0.0001), and specifically for oncologic patients’ ED visits (odds = 0.43, *p* < 0.0001). In 2021, oncologic patients’ ED visits increased back to the numbers seen in 2019 (odds = 0.94, *p* = 0.16, Table 1), while total ED visits also increased but not to the same level as in 2019 (odds = 0.65, *p* < 0.0001).

As shown in Table 2, no statistically significant differences were reported for gender, race, or age.

The decrease and subsequent increase in 2020 and 2021, respectively, compared to 2019 were similar for almost all tumor sites and for patients in curative/palliative treatment or off treatment (Figure 1, Table 2).

In fact, according to the multivariate analysis, as evidenced by the forest plot (Figure 1A), brain, melanoma, and sarcoma tumors showed a reduction from 2019 to 2020, while gynecological tumors increased. Head and neck (HN), gastrointestinal (GI), genitourinary (GU), hematological, lung, and breast tumors remained stable (Figure 1A).

By 2021, all tumors returned to 2019 levels, except for gynecological ones, which increased instead (Figure 1B and Figure 2).

Interestingly, as shown in Figure 3, the proportion of patients with metastatic disease increased in 2020 (22.8% vs. 15.5% in 2019) before decreasing again in 2021 (13.8%). Meanwhile, those with locally advanced disease had a reduction in ED visits in 2020 (24.9% vs. 33.5% in 2019; *p* = 0.020) and returned to 36.8% in 2021 (*p* = 0.30). Patients with early-stage disease were found to have similar trends across the three years (51.1% in 2019, 52.2% in 2020, and 49.3% in 2021).

Using the Charlson index, we observed a dramatic decrease in ED visits for those with a score of >9 points in 2019 (19.0% vs. 4.4% in 2020; *p* < 0.0001) and in 2021 (0.9%; *p* < 0.0001).

ED visits’ characteristics, including severity code, ED specialty area, method of arrival, and reason for visit, are described in Table 3. A sharp increase was observed in 2020 compared to 2019 for ED visits due to cardiac symptoms (6.6% vs. 7.4%) and bleeding (8.9% vs. 9.6%), while organic failure (6.1% vs. 3.1%), device obstruction (6.0% vs. 4.6%)**,** trauma (7.0% vs. 4.8%), and pain (20.5% vs. 14%) decreased. ED visits by patients presenting with COVID-19-like symptoms such as fever and dyspnea increased significantly in 2020 compared to 2019 (fever 15.5% vs. 11.8% and dyspnea 14.4% vs. 8.8%; *p* = 0.001).

Finally, the percentage of each severity code (red, yellow, green, and white) remained stable across the three years. Concerning the specialty areas within the ED, a reduction was seen in 2020 for the medicine department (54.9% vs. 23.1% in 2019), as well as for surgery (32% vs. 27.2%) and orthopedics (4.7% vs. 2.6%; *p* < 0.001) before returning to normalcy in 2021 (*p* = 0.15).

The percentage of discharged patients was reduced by about one-third in 2020 (62.7% vs. 44.0% in 2019). At the same time, 48.6% of patients were admitted in 2020, an increase from 35.0% in 2019. The outcome of death doubled in 2020 (2.6% vs. 1.2% in 2019), while voluntary discharge tripled (1.1% vs. 3.1%; *p* < 0.0001).

Among those who were admitted to the hospital, an almost two-fold increase was seen for those admitted to surgery in 2020 compared to 2019 (11.7% vs. 20.7%). Meanwhile, reductions in admissions were noted in 2020 compared to 2019 for those admitted to neurosurgery (6.3% vs. 0.9%), orthopedic surgery (1.6% vs. 0.5%), internal medicine (16.3% vs. 6.3%), and hematology (4.3% vs. 0.9%; *p* < 0.0001). The proportion of patients admitted to the oncology department remained the same (13.0% in 2019 vs. 12.2% in 2020). 

ED visits by ambulance increased from 2019 to 2020 (29.1% vs. 44.0%). Visits by patients arriving by car, on the other hand, were reduced (66.8% vs. 53.6%; *p* < 0.0001), before returning again in 2021 (67.3%; *p* = 0.11). Finally, ED visits by COVID-positive patients greatly decreased in 2021 compared to 2020 (32.2% vs. 7.6%; *p* < 0.0001).

## 4. Discussion

Liang et al. in 2020 noted that oncologic patients developed more severe events (a composite endpoint that the authors defined as the percentage of patients being admitted to the intensive care unit requiring invasive ventilation, or death) compared to non-oncologic patients (39% vs. 8%, *p* = 0.0003) [12]. As various aspects of patient care were postponed or moved towards remote delivery, this was not an acceptable reality for many cancer patients. Based on severity, patients’ physical presence at hospitals was required, putting them at high risk of COVID-19 exposure. Moreover, given the immunosuppressive nature of many oncologic treatments, patients were again at increased risk of infection and the development of severe disease [21]. 

Institutes providing cancer care rapidly implemented infection-control rules. This ultimately meant separating or limiting family members and friends from accompanying patients to their appointments or visiting them when hospitalized. Treatment was delayed in some places, resulting in patients and their families worrying about the consequential impact on their outcomes [22]. In fact, during the pandemic, anxiety prevalence among patients was 19.1%, 22.5% among caregivers, and 14.0% among healthcare workers (*p* = 0.004). Of note, anxiety was higher in those without post-secondary education (OR, 1.78; 95% CI, 1.04 to 3.15; *p* = 0.04) and those who were married (OR, 2.11; 95% CI, 1.14 to 4.22; *p* = 0.025). Fear, followed by anxiety, were reported as the two most frequent emotions experienced by patients. Healthcare workers were found to be less fearful (41.6%) compared to patients and caregivers (66.0% and 72.8%, respectively) (*p* = 0.001) [18]. Caregivers experienced more fear than patients with regards to how the pandemic may affect cancer outcomes (72.1% v 54.5%; *p* = 0.001). The top COVID-19-related fears experienced by patients, caregivers, and healthcare workers were the risk of mass community spread, spending their potentially last hours alone, and exhibiting COVID-19 symptoms that were too mild to be recognized on time [20]. Therefore, fear could have been a reason behind our findings of reduced oncologic patients’ ED visits during the pandemic. Patients and caregivers expressed high confidence in healthcare workers to recognize COVID-19 symptoms and in healthcare facilities to manage the COVID-19 outbreak [20].

Tabriz et al., in a cross-sectional study, reported that 51.6% of ED visits made by cancer patients are potentially preventable, since the main reasons behind the visits are pain, fever, nausea, and vomiting; while other conditions, such as dyspnea, fatigue, urinary tract infections, syncope, dizziness, giddiness, and acute exacerbation of obstructive pulmonary disease, were not preventable reasons for ED visits [23]. 

During the pandemic, many measures were adopted by the Italian healthcare system for the regulation of outpatient management with the aim to reduce hospital admissions and protect non-COVID-19 patients. Following the development of the World Health Organization (WHO)’s international recommendations, several guidelines were published in Italy, and more specifically, within Lombardy (https://www.ats-brescia.it/disposizioni-ufficiali accessed on 1 March 2020) on how to manage those with suspected and confirmed positive molecular assessments of SARS-CoV-2 [24,25]. 

During the pandemic, cancer health professionals faced a rapid need to make decisions about how to best manage their patients. The threat of a health system overwhelmed by COVID-19 patients was a valid concern. Moreover, general pandemic measures such as social distancing, along with the uncertainties surrounding COVID-19, affected healthcare professionals no less than the general population [26]. Consequently, COVID-19 had an impact on both recipients and providers of oncologic care [22].

From an oncological point of view, our hospital strived towards delivering optimal treatments for our cancer patients while simultaneously decreasing the risk of COVID-19 exposure and infection. This was a difficult equation to balance, as almost all resources were redirected to serve COVID-19 patients. Nonetheless, although ambulatory activities were discouraged, our hospital’s mission to treat cancer patients was maintained thanks to the immediate implementation of the necessary measures.

At an institutional level, screening upon entry was set up, preventing entry to subjects with a body temperature over 37.5 °C and limiting caregiver access to select cases. PPE and biweekly COVID-19 testing were provided for healthcare workers. The use of hand sanitizers and surgical masks was mandated within the hospital. Seating areas were restricted to ensure social distancing. Moreover, crowded waiting rooms were avoided through the creation of an online system that allowed patients to obtain a reservation number in advance.

Our hospital’s ED was reorganized: Its size was enlarged to create an area dedicated to COVID-positive patients, suitable to satisfy a large number of these visits. Medical and nursing staff were reassigned to cover either COVID or non-COVID areas. Moreover, most surgical and medical departments were converted into COVID departments except cardiology, stroke unit, cardiac surgery, intensive therapy, neurosurgery, a minimal part of internal medicine, and about half of the medical oncology department. These are possible explanations for the varying numbers of hospital admissions per department. For example, the reduction in internal medicine and hematology wards may have possibly been due to their partial conversion to COVID wards along with the less acute nature of their disease, allowing for management at home. An increase rather than a decrease was observed in surgery, likely due to urgent pathologies that could not be managed at home. The closure of half of the oncology department and part of the internal medicine department likely had a negative impact on patient admission into these departments from the ED, further complicating management for oncologic patients.

The increase in ED visits by patients with metastatic disease in 2020 could be explained by the nature of their advanced disease, for example, symptoms severe enough and time-sensitive where at-home management is not sufficient. On the other hand, patients with locally advanced disease visited the ED less in 2020, possibly due to hopeful waiting for better circumstances, given the uncertainty at the time. Furthermore, we observed a dramatic decrease in ED visits by patients with higher Charlson index scores, meaning fewer patients with a high number of comorbidities visited the ED in 2020 compared to 2019.

ED visits based on primary tumor sites were generally stable, except for melanoma, sarcoma, and brain. The reduction in ED visits by melanoma and sarcoma patients can be explained by mere chance, given the very low number of cases (<20 patients in the reference year, 2019) and the upper limit of the 95% OR CI being above the unit value, in both the univariate and multivariate analyses. Visits by patients with brain tumors decreased from 2019 (5.8%) to 2020 (2.6%) (OR: 0.42; 95% CI 0.20–0.89), before increasing in 2021 (3.1%) (OR: 0.55; 95% CI 0.31–0.98), likely as a result of our institution becoming a referral hub for neurosurgical diseases in 2020, and thus patients were directly admitted into the neurosurgery department bypassing the ED. While visits by patients with gynecological tumors slightly increased in 2020, also explained by chance, we noted an extraordinary increase in 2021, potentially due to the reorganization of our hospital during the pandemic.

An important limitation of this analysis is the lack of quality-of-life data since this is a retrospective study, along with the difficulty of administering such questionnaires in an ED context. Other factors to consider include varying primary care contributions within the population and cultural differences between Italians and those of other ethnicities, as these likely had an impact on ED visits, patient behavior, and their sense of vulnerability. Nonetheless, trends were similar to results from a retrospective cohort study in Canada studying the impact of the pandemic on ED visits for patients undergoing cancer-directed surgery, as well as studies on ED visits by non-oncologic patients in the USA and in Israel [27,28,29].

Primary care in Italy is not without difficulties: As the Italian National Health Service is managed regionally, different regions of the country experienced large variations in COVID-19 management. The response to the pandemic in Lombardy specifically came with significant shortcomings [30,31] and required better coordination between primary care doctors, public health experts, social services, and community organizations to have a more effective impact.

During the first wave of the pandemic in 2020, the Italian government adopted many measures to reduce the risk of contagion, such as suspending common retail commercial activities, educational activities, and catering services, all with the aim to curb the gathering of people in public places. Leaving one’s domicile was also limited to basic necessities (i.e., grocery shopping, medical attention). All these factors could have contributed to the reduction in ED visits by cancer patients in 2020. Thanks to the kickoff of the national vaccination campaign in 2021, many of these restrictive measures were gradually lifted, though wearing a mask was still mandated. 

In this context, people in 2021 felt more protected against the virus and were more comfortable going out, resuming their normal activities, and visiting the ED more liberally.

## 5. Conclusions

During the pandemic, many resources and efforts were dedicated to SARS-CoV-2 management, potentially neglecting other critical fields of medicine such as oncology. To our best knowledge, this is the first Italian study analyzing the problem from the perspective of patients’ ED visits. Our data demonstrate a significant decrease in ED visits during the pandemic in 2020 compared to 2019.

Based on this data, we can speculate that fear, anxiety, and worry, in addition to hospital departmental reorganization, negatively influenced oncological outcomes, both in terms of treatment benefits and psychological well-being. Moreover, even in the presence of symptoms, delayed presentation to medical care compromises early diagnosis, and in some cases, cure, while increasing the chances of developing metastatic disease in many patients.

## Figures and Tables

**Figure 1 cancers-15-01240-f001:**
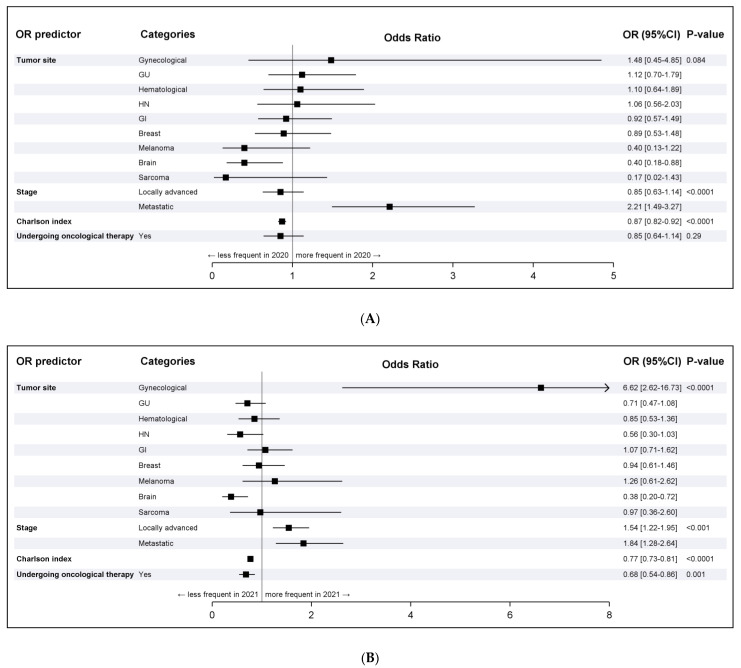
Multivariate analysis. (**A**) 2020 vs. 2019. (**B**) 2021 vs. 2019. **Note:** OR estimates were obtained using the following reference levels: Tumor site: lung; Stage: early disease; In active oncological therapy?: no. The comparison was performed between 2019 and 2020. **Note:** OR estimates were obtained using the following reference levels: Tumor site: lung; Stage: early disease; In active oncological therapy?: no. The comparison was performed between 2019 and 2021.

**Figure 2 cancers-15-01240-f002:**
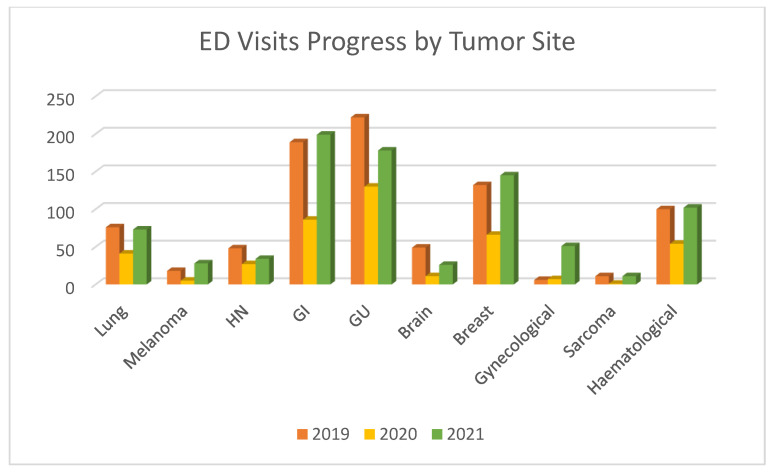
ED visits progress by tumor primary site.

**Figure 3 cancers-15-01240-f003:**
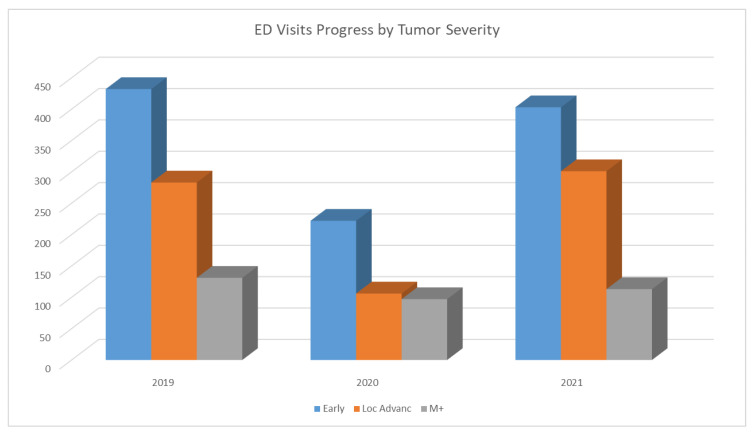
ED visits progress by tumor severity.

**Table 1 cancers-15-01240-t001:** ED oncological visits.

Year	First Access	Last Access ^§^	N° of Accesses			N° of Patients		
			Total (%)	Per Week	Odds (*p*-Value)	Total (%) **	Per Week	Odds (*p*-Value)
2019	7 March 2019	17 May 2019	1061 (42.2)	103.2	1	893 (42.0)	86.8	1
2020	7 March 2020	17 May 2020	459 (18.2)	44.6	0.43 (<0.0001)	432 (20.3)	42.0	0.48 (<0.0001)
2021	7 March 2021	17 May 2021	996 (39.6)	96.8	0.94 (0.16)	848 (39.9)	82.4	0.95 (0.28)
**Overall**			**2516 (100)**	**81.5**	**-**	**2125**	**68.9**	**-**

^§^ The time interval of ED visits was 72 days for each year. ** The total number of patients is less than the sum of the patients per year because there are some patients visited the ED more than once in different years.

**Table 2 cancers-15-01240-t002:** Patient’s characteristics.

Characteristics	Category	#	2019	2020	Odds Ratio (95%CI) ^1^	*p*-Value	2021	Odds Ratio (95%CI) ^2^	*p*-Value
Gender	Male	N	477	238	1	0.57	449	1	0.85
		%	53.4	55.1			52.9		
	Female	N	416	194	0.93 (0.74–1.18)		399	1.02 (0.84–1.23)	
		%	46.6	44.9			47.1		
Age (years)	-	N	889	432	1.07 (0.99–1.16) *	0.098	848	1.06 (0.996–1.13) *	0.066
		Median	71.2	72.5			73.0		
		Min–max	11.5–97.6	32.4–96.1			4.2–96.5		
Ethnicity	Caucasian	N	872	422	1	0.93	814	1	0.048
		%	97.8	97.7			96.1		
	Others	N	20	10	1.03 (0.48–2.23)		33	1.77 (1.01–3.11)	
		%	2.2	2.3			3.9		
	*Missing data*	*N*	*1*	*0*			*1*		
		*%*	*0.1*	*0*			*0.1*		
Tumor site	Lung	N	76	41	1	0.087	73	1	<0.0001
		%	8.9	9.6			8.6		
	Melanoma	N	18	5	0.51 (0.18–1.49)		28	1.62 (0.83–3.18)	
		%	2.1	1.2			3.3		
	HN	N	48	27	1.04 (0.57–1.91)		34	0.74 (0.43–1.27)	
		%	5.6	6.3			4.0		
	GI	N	189	86	0.84 (0.53–1.33)		199	1.10 (0.75–1.60)	
		%	22.2	20.1			23.5		
	GU	N	222	130	1.09 (0.70–1.68)		178	0.83 (0.57–1.22)	
		%	26.1	30.4			21.0		
	Brain	N	49	11	0.42 (0.20–0.89)		26	0.55 (0.31–0.98)	
		%	5.8	2.6			3.1		
	Breast	N	132	66	0.93 (0.57–1.50)		145	1.14 (0.77–1.70)	
		%	15.5	15.4			17.1		
	Gynecological	N	6	7	2.16 (0.68–6.86)		51	8.85 (3.58–21.87)	
		%	0.7	1.6			6.0		
	Sarcoma	N	11	1	0.17 (0.02–1.35)		11	1.04 (0.43–2.55)	
		%	1.3	0.2			1.3		
	Hematological	N	100	54	1.00 (0.60–1.66)		102	1.06 (0.70–1.62)	
		%	11.8	12.6			12.0		
	*Missing data*	*N*	*42*	*4*			*1*		
		*%*	*4.7*	*0.9*			*0.1*		
Stage	Early	N	432	222	1	0.020	403	1	0.30
		%	51.1	52.2			49.3		
	Loc. advanc.	N	283	106	0.73 (0.55–0.96)		301	1.14 (0.92–1.41)	
		%	33.5	24.9			36.8		
	M+	N	131	97	1.44 (1.06–1.96)		113	0.92 (0.69–1.23)	
		%	15.5	22.8			13.8		
	*Missing data*	*N*	*47*	*7*			*31*		
		*%*	*5.3*	*1.6*			*3.7*		
Disease duration (year)	*-*	N	893	432	1.01 (0.85–1.21) *	0.89	848	1.12 (0.97–1.28) *	0.12
		Median	2.7	2.8			3.7		
		Min–max	0.5–48.7	0.5–37.8			0.5–47.8		
Charlson index	≤5	N	270	142	0.90 (0.86–0.95)	<0.0001	346	0.79 (0.76–0.83)	<0.0001
		%	30.2	32.9			40.8		
	6–9	N	453	271			494		
		%	50.7	62.7			58.3		
	>9	N	170	19			8		
		%	19.0	4.4			0.9		
In oncological therapy?	No	N	412	213	1	0.82	449	1	0.008
		%	49.0	49.7			55.6		
	Yes	N	429	216	0.97 (0.77–1.23)		359	0.77 (0.63–0.93)	
		%	51.0	50.3			44.4		
	*Missing data*	*N*	*52*	*3*			*40*		
		*%*	*5.8*	*0.7*			*4.7*		
Type of final oncological therapy	Curative	N	733	330	1	0.066	725	1	0.78
		%	85.5	81.5			86.0		
	Palliative	N	124	75	1.34 (0.98–1.84)		118	0.96 (0.73–1.26)	
		%	14.5	18.5			14.0		
	*Missing data*	*N*	*36*	*27*			*5*		
		*%*	*4.0*	*6.2*			*0.6*		

^#^ Absolute and percentage frequencies for categorical variables and absolute frequencies, median, and min–max values for continuous variables are reported. **^1^** Odds ratio estimated between 2019 and 2020 years **^2^** Odds ratio estimated between 2019 and 2021 years. * Odds ratio estimated for 10 years interval.

**Table 3 cancers-15-01240-t003:** Characteristics of ED oncological accesses.

Characteristics	Category	#	2019	2020	Odds Ratio (95%CI) ^1^	*p*-Value	2021	Odds Ratio (95%CI) ^2^	*p*-Value
Severity code	Red	N	75	31	1	0.79	59	1	0.68
		%	7.1	6.8			5.9		
	Yellow	N	380	153	0.97 (0.62–1.54)		348	1.16 (0.80–1.69)	
		%	35.8	33.3			34.9		
	Green	N	600	272	1.10 (0.70–1.71)		583	1.24 (0.86–1.77)	
		%	56.6	59.3			58.5		
	White	N	6	3	1.21 (0.28–5.15)		6	1.27 (0.39–4.14)	
		%	0.6	0.7			0.6		
Access area	Medicine	N	582	106	1	<0.0001	436	1	0.15
		%	54.9	23.1			43.8		
	Surgery	N	349	125	1.97 (1.47–2.63)		337	1.29 (1.06–1.57)	
		%	32.9	27.2			33.8		
	COVID	N	0	147	nd		76	nd	
		%	0	32.0			7.6		
	Orthopedic	N	50	12	1.32 (0.68–2.56)		50	1.33 (0.88–2.01)	
		%	4.7	2.6			5.0		
	Gynecology	N	31	10	1.77 (0.84–3.72)		20	0.86 (0.48–1.53)	
		%	2.9	2.2			2.0		
	Emergency	N	26	6	1.27 (0.51–3.15)		12	0.62 (0.31–1.23)	
		%	2.5	1.3			1.2		
	Oculists	N	18	4	1.22 (0.40–3.68)		12	0.89 (0.42–1.87)	
		%	1.7	0.9			1.2		
	Pediatric	N	5	0	0 (0-nd)		2	0.53 (0.10–2.77)	
		%	0.5	0			0.2		
	Other	N	0	2	nd		0	nd	
		%	0	0.4			0		
Method of arrival	Ambulance	N	309	202	1	<0.0001	302	1	0.11
		%	29.1	44.0			30.3		
	Surgery/DH	N	43	11	0.39 (0.20–0.78)		24	0.57 (0.34–0.96)	
		%	4.1	2.4			2.4		
	Personal transportation	N	709	246	0.53 (0.42–0.67)		670	0.97 (0.80–1.17)	
		%	66.8	53.6			67.3		
Cause for visit	Pain	N	218	64	1	0.001	198	1	0.50
		%	20.5	14.0			19.9		
	Neurological symptoms	N	143	65	1.55 (1.03–2.32)		126	0.97 (0.71–1.32)	
		%	13.5	14.2			12.7		
	Fever	N	125	71	1.93 (1.29–2.90)		107	0.94 (0.68–1.30)	
		%	11.8	15.5			10.7		
	Dyspnea	N	93	66	2.42 (1.59–3.68)		108	1.28 (0.91–1.79)	
		%	8.8	14.4			10.8		
	Bleeding	N	94	44	1.59 (1.01–2.51)		101	1.18 (0.84–1.66)	
		%	8.9	9.6			10.1		
	Vomit/GI symptoms	N	84	39	1.58 (0.99–2.53)		76	1.00 (0.69–1.44)	
		%	7.9	8.5			7.6		
	Organ failure	N	65	14	0.73 (0.39–1.39)		80	1.36 (0.93–1.98)	
		%	6.1	3.1			8.0		
	Cardiac symptoms	N	70	34	1.65 (1.01–2.72)		51	0.80 (0.53–1.21)	
		%	6.6	7.4			5.1		
	Devices Obstructions	N	64	21	1.12 (0.63–1.97)		68	1.17 (0.79–1.73)	
		%	6.0	4.6			6.8		
	Trauma	N	74	22	1.01 (0.58–1.76)		54	0.80 (0.54–1.20)	
		%	7.0	4.8			5.4		
	Jaundice	N	17	11	2.20 (0.98–4.94)		15	0.97 (0.47–2.00)	
		%	1.6	2.4			1.5		
	Hypotension	N	6	3	1.70 (0.41–7.00)		10	1.84 (0.65–5.14)	
		%	0.6	0.7			1.0		
	Pneumonia	N	2	4	6.81 (1.22–38.05)		2	1.10 (0.15–7.89)	
		%	0.2	0.9			0.2		
	Other	N	6	0	0 (0-nd)		0	0 (0-nd)	
		%	0.6	0			0		
	*Missing data*	*N*	*0*	*1*			*0*		
		*%*	*0*	*0.2*			*0*		
Duration of stay (days)	-	N	1056	448	1.33 (1.15–1.54)	0.0001	996	1.40 (1.26–1.56)	<0.0001
		Median	0.39	1.00			1.00		
		Min–max	0.003–5.00	0.004–30.27			0.007–6.00		
Outcome	Discharge	N	665	202	1	<0.0001	603	1	0.52
		%	62.7	44.0			60.5		
	Admission	N	371	223	1.98 (1.57–2.49)		368	1.09 (0.91–1.31)	
		%	35.0	48.6			36.9		
	Voluntary discharge	N	12	14	3.84 (1.75–8.44)		16	1.47 (0.69–3.13)	
		%	1.1	3.1			1.6		
	Death	N	13	12	3.04 (1.37–6.76)		9	0.76 (0.32–1.80)	
		%	1.2	2.6			0.9		
	Moved to another hospital	N	0	8	nd		0	nd	
		%	0	1.7			0		
Hospitalization ward	Dimer	N	68	16	1	<0.0001 *	57	1	<0.0001 *
		%	18.5	7.2			15.5		
	Oncology	N	48	27	2.39 (1.16–4.91)		53	1.32 (0.78–2.23)	
		%	13.0	12.2			14.4		
	Surgery	N	43	46	4.55 (2.29–9.02)		33	0.92 (0.52–1.63)	
		%	11.7	20.7			9.0		
	Internal medicine	N	60	14	0.99 (0.45–2.20)		47	0.93 (0.56–1.57)	
		%	16.3	6.3			12.8		
	COVID	N	0	57	nd		56	nd	
		%	0	25.7			15.2		
	Urology	N	23	17	3.14 (1.37–7.21)		45	2.33 (1.26–4.31)	
		%	6.3	7.7			12.2		
	Neurology	N	27	12	1.89 (0.79–4.51)		22	0.97 (0.50–1.89)	
		%	7.3	5.4			6.0		
	Cardiology	N	18	14	3.31 (1.36–8.01)		24	1.59 (0.79–3.22)	
		%	4.9	6.3			6.5		
	Neurosurgery	N	23	2	0.37 (0.08–1.73)		8	0.41 (0.17–1.00)	
		%	6.3	0.9			2.2		
	Gynecology	N	16	6	1.59 (0.54–4.72)		6	0.45 (0.16–1.22)	
		%	4.3	2.7			1.6		
	Hematology	N	16	2	0.53 (0.11–2.55)		2	0.15 (0.03–0.68)	
		%	4.3	0.9			0.5		
	Orthopedic	N	6	1	0.71 (0.08–6.30)		8	1.59 (0.52–4.85)	
		%	1.6	0.5			2.2		
	UTI/UTIC	N	4	5	5.31 (1.28–22.05)		5	1.49 (0.38–5.82)	
		%	1.1	2.3			1.4		
	Infectious Diseases	N	11	0	0 (0-nd)		0	0 (0-nd)	
		%	3.0	0			0		
	ORL	N	1	0	0 (0-nd)		2	2.39 (0.21–27.00)	
		%	0.3	0			0.5		
	Other	N	4	3	3.19 (0.65–15.68)		0	0 (0-nd)	
		%	1.1	1.4			0		
	*Missing data*	N	*3*	*1*			*0*		
		%	*0.8*	*0.4*			*0*		
COVID	Negative	N	1061	311	1	<0.0001 *	920	1	<0.0001 *
		%	100.0	67.8			92.4		
	Positive	N	0	148	nd		76	nd	
		%	0	32.2			7.6		

^#^ Absolute and percentage frequencies for categorical variables and absolute frequencies, median and min–max values for continuous variables are reported. **^1^** Odds ratio estimated between 2019 and 2020 years **^2^** Odds ratio estimated between 2019 and 2021 years. * Odds ratio estimated for 10 years interval. * An exact logistic regression model was used.

## Data Availability

No new data have been created.

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
