# Peer review of "Emergency Department Visits among Cancer Patients during SARS-CoV-2 Pandemic"

_cancers, 2023, doi:10.3390/cancers15041240_

Round 1

Reviewer 1 Report

This is a single, tertiary level, cancer centre-based study in one of Italy's hardest hit regions from COVID-19 - comparing/contrasting ED cancer patient admissions before during and later in the COVID-19 pandemic.

Such studies are very unique to the region and population, so as a descriptive study it is useful at a general level, but difficult to compare across different countries/population/institutions, as the cancer patient mix varies enormously, as well as the treatment protocols.

But also not considered by the authors, are the other factors that might impact on the ED admissions:

- the degree to which primary care (and any related community healthcare support) was impacted COVID - in some countries there is extensive community support or vulnerable people without the need to be hospitalised (e.g. UK) but this is much less in other countries (e.g. USA)

- the weather and its impact - as colder weather may impact on such vulnerable patients more adversely than warmer weather - but if the weather becomes too warm, this can also impact on their well-being.

- cultural differences - Italians are very gregarious, sociable people and may not handled isolation during lockdown than, say, Scandinavian populations, who are more used to isolation and cold, harsh weather than populations from Mediterranean regions - and this can impact on the ED attendances and patient behaviour and sense of vulnerability.

The authors should also report, in parallel, the different national COVID-19 restrictions as well as the introduction of vaccines in 2021 - and how these impacted on the ED attendance - so a timeline of these COVID-19 restrictions and how the changed from 2019 (none), to 2020 (severe - give details), to 2021 (milder in the context of vaccination - give details) would enable readers to put the ED attendance trends in context - and allow them to extrapolate the potential impact to their own populations - with the limitations of the other local factors/caveats, such as those listed above.

Some comments on these issues as related to the generalisation of these results, by the authors, would be useful for readers.

Otherwise, the overall findings are not surprising given the early fears about COVID-19.

Author Response

Thank you for your interest in our manuscript and for your precious suggestions, which have greatly helped improve the manuscript.

An analyses of primary care and cultural differences impact have been added at the end of the discussion.

Less influence had the weather since in Italy from march to may it's warm. This it the reason why we did not considered it in the paper.

National restrictions have been added as well.

Reviewer 2 Report

Interesting topic but the manuscript need a lot correction work:

1. Extensive editing of English language and style required

2. Introduction section

Line 33 - Compared to the general population, oncologic patients are at a higher risk of worse outcomes due to infectious diseases – missing reference

Line 67-68 - One potential consequence of this emotional distress is a decrease in ED visits among oncologic patients, which to our knowledge, has not been reported in the literature. – check this reference: ED utilization trends for medical oncology patients at Thomas Jefferson University during COVID-19.Zachary L. Quinn, Adam Binder, Valerie Pracilio Csik, Helen Evers-Hunt, and Nathan HandleyJournal of Clinical Oncology 2020 38:29_suppl, 238-238

Materials and methods

Line 81 - We considered at least a 20% decrease in ED visits to be clinically significant.  – need more explanation

Discussion section

Line 190-191: Oncologic patients are reported to develop more severe events such as ICU admis-190 sion, invasive ventilation or death, compared to non-oncologic patients (39% vs 8%, p 191 =0.0003) – in term of what?

Missing references

Zachary L. Quinn, Adam Binder, Valerie Pracilio Csik, Helen Evers-Hunt, and Nathan Handley.ED utilization trends for medical oncology patients at Thomas Jefferson University during COVID-19. Journal of Clinical Oncology 2020 38:29_suppl, 238-238

Majka ES, Trueger NS. Emergency Department Visits Among Patients With Cancer in the US. JAMA Netw Open. 2023;6(1):e2253797. doi:10.1001/jamanetworkopen.2022.53797

Eskander A, Li Q, Yu J, Hallet J, Coburn N, Dare A, Chan KKW, Singh S, Parmar A, Earle CC, Lapointe-Shaw L, Krzyzanowska MK, Hanna TP, Finelli A, Louie AV, Look-Hong N, Irish JC, Witterick I, Mahar A, Urbach DR, Enepekides D, Sutradhar R, On Behalf Of The Pandemic-Ontario Collaborative In Cancer Research Poccr. Assessing the Impact of the COVID-19 Pandemic on Emergency Department Use for Patients Undergoing Cancer-Directed Surgeries. Curr Oncol. 2022 Mar 10;29(3):1877-1889.

Sagy YW, Cicurel A, Battat E, Saliba W, Lavie G. The impact of COVID-19 pandemic on emergency department visits and associated mortality during 14 months of the pandemic in Israel. Intern Emerg Med. 2022 Sep;17(6):1699-1710.

Kram DE, Tooze JA, Russell TB, McLean TW. COVID-19-Related Reduction in Emergency Health Care Utilization Among Febrile Pediatric Oncology Patients. J Pediatr Hematol Oncol. 2022 Apr 1;44(3):e649-e652.

Author Response

Thank you for your interest in our manuscript and for your precious suggestions, which have greatly helped improve the manuscript.

  1. Extensive editing of English language and style required. One of the author is an English mother tongue, but we have proceeded to review it again.
  2. Introduction section

Line 33 - Compared to the general population, oncologic patients are at a higher risk of worse outcomes due to infectious diseases – missing reference. We have now added the reference to the text

Line 67-68 - One potential consequence of this emotional distress is a decrease in ED visits among oncologic patients, which to our knowledge, has not been reported in the literature. – check this reference: ED utilization trends for medical oncology patients at Thomas Jefferson University during COVID-19.Zachary L. Quinn, Adam Binder, Valerie Pracilio Csik, Helen Evers-Hunt, and Nathan HandleyJournal of Clinical Oncology 2020 38:29_suppl, 238-238. We have now added this information and the reference to the text.

Materials and methods

Line 81 - We considered at least a 20% decrease in ED visits to be clinically significant.  – need more explanation. We removed this information since it was only a clinical consideration, not useful for the paper.

Discussion section

Line 190-191: Oncologic patients are reported to develop more severe events such as ICU admis-190 sion, invasive ventilation or death, compared to non-oncologic patients (39% vs 8%, p 191 =0.0003) – in term of what? We have now explained this point in the text.

Missing references. We have now added all the references to the text except for the last one because it was about pediatrics, excluded from our analyses.

Reviewer 3 Report

Authors conducted a study on emergency department admission for cancer patients during the COVID-19 pandemic and found out that there is a significant decrease in ED visits. 

'Statistics' in table headers should be changed to some other appropriate word. There are so many paragraphs in small number of sentences. Authors need to properly improve the writing. 

Moreover, author needs to address some of the large differences in discussion such as ED visit drops in brain and sarcoma patients. Also, why patients with higher Charlston index declined. 

Author Response

Thank you for your interest in our manuscript and for your suggestions.

'Statistics' in table headers should be changed to some other appropriate word. We have now changed the text as requested.

There are so many paragraphs in small number of sentences. Authors need to properly improve the writing. We have now rephrased the paragraphs following as suggested.

Moreover, author needs to address some of the large differences in discussion such as ED visit drops in brain and sarcoma patients. We have now added these informations to the text

Also, why patients with higher Charlston index declined. We have now added these informations to the text

Round 2

Reviewer 2 Report

The authors made requested changes.